# Photocatalyzed cycloaromatization of vinylsilanes with arylsulfonylazides

Fengjuan Chen[1,4], Youxiang Shao[2,4], Mengke Li[3,4], Can Yang[1], Shi-Jian Su [3✉], Huanfeng Jiang[1], Zhuofeng Ke [2✉] & Wei Zeng [1✉]

Sila-molecules have recently attracted attention due to their promising applications in medical and industrial fields. Compared with all-carbon parent compounds, the different covalent radius and electronegativity of silicon from carbon generally endow the corresponding sila-analogs with unique biological activity and physicochemical properties. Vinylsilanes feature both silyl-hyperconjugation effect and versatile reactivities, developing vinylsilane-based Smiles rearrangement will therefore provide an efficient platform to assemble complex silacycles. Here we report a practical Ir(III)-catalyzed cycloaromatization of *ortho*-alkynylaryl vinylsilanes with arylsulfonyl azides for delivering naphthyl-fused benzosiloles under visible-light photoredox conditions. The combination of experiments and density functional theory (DFT) energy profiles reveals the reaction mechanism involving α-silyl radical Smiles rearrangement.

[1] Key Laboratory of Functional Molecular Engineering of Guangdong Province, School of Chemistry and Chemical Engineering, South China University of Technology, Guangzhou, China. [2] School of Materials Science and Engineering, PCFM Lab, Sun Yat-sen University, Guangzhou, China. [3] State Key Laboratory of Luminescent Materials and Devices, Institute of Polymer Optoelectronic Materials and Devices, South China University of Technology, Guangzhou, China. [4]These authors contributed equally: Fengjuan Chen, Youxiang Shao, Mengke Li. ✉email: mssjsu@scut.edu.cn; kezhf3@mail.sysu.edu.cn; zengwei@scut.edu.cn

Silahydrocarbons are sometimes encountered in pharmaceuticals and material chemistry[1–4]. Compared with all-carbon parent compounds, Si-element generally endows the corresponding hydrocarbons with unique biological activity and physical–chemical properties[5–8], which are mainly determined by the different covalent radius and electronegativity of silicon from carbon. In these regards, arene-fused siloles have especially attracted many concerns due to their promising applications in electronic and optoelectronic devices[9–13]. In 2012, Xi[14] and Chatani[15] pioneeringly explored intermolecular coupling-cyclization of alkynes with 2-silylaryl bromides and 2-silylphenylboronic acids to produce 2,3-difunctionalized benzosiloles through Si-C bond cleavage (Fig. 1a). Subsequently, He[16] developed rhodium-catalyzed intramolecular silicoamination of ortho-alkynylarylsilanes to construct indole-fused benzosiloles (Fig. 1b). In light of that modifying the π-conjugated system of parent siloles could possibly improve their corresponding photophysical properties. Thus, there has been an ever-increasing demand for the rapid assembly of diversified polycycle-fused siloles[17–22].

Aryl migration via Smiles rearrangement is a powerful tool for the synthesis of polycyclic arenes[23–25]. However, the modes of radical Smiles rearrangement are very limited. Up to now, only α-carbonyl radical[26–30], β-aminoalkyl radical[31–33], N-centered radical[34], and ketyl radical[35]- triggered Smiles rearrangement have been exploited to construct nitrogen-heterocycles, and vinylsilane-based Smiles rearrangement keep unexplored. As is well known, vinylsilanes have proven to be important "alkene" sources in Hiyama coupling, which could efficiently incorporate C=C bond in a particular molecule with the release of silyl moiety[36]; Meanwhile, the high electronegativity of carbon (2.35) relative to silicon (1.64)[37] and silyl-hyperconjugation effect (the so-called β-effect)[38–40] generally endow these compounds with the versatile reactivity. For example, Jun[41] reported that Rh(I)-catalyzed cross-coupling of aldehydes with vinylsilanes led to the formation of β-acylsilanes via β-silylethylrhodium(III) intermediates. On the contrary, Buchwald[42] and Miura[43] demonstrated that Cu(I)-catalyzed addition-coupling of vinylsilanes with amines could produce α-aminosilanes. Thus it can be seen that

developing vinylsilane-based coupling-cyclization will possibly establish an efficient platform to assemble complex silacycles. Again, azides could be employed as potential nitrogen radical precursors to enable C-H amination[44] under photocatalysis systems. Accordingly, visible-light catalyzed coupling of vinylsilanes with arylsulfonylazides could possibly generate α-silyl radicals and initiate the silylalkene Smiles rearrangement.

Here, we show a cycloaromatization of ortho-alkynylarylsilylalkenes with arylsulfonyl azides for rapid assembly of 2,3-naphthyl-fused benzosiloles via a cascade S-N/C-S bond cleavage in the presence of visible light (Fig. 1c).

## Results

**Investigation of reaction conditions.** The choice of ortho-alkynylaryl vinylsilane **1a** was motivated by the notion that alkynyl and vinyl groups could possibly trap radicals to assemble complex silacycles under blue light-emitting diodes (LEDs) irradiation. After an extensive screening of various reaction parameters, we were pleased to find that the treatment of substrate **1a** with TsN₃ **2a** under photocatalyzed system did afford a large π-conjugated benzosilafluorene **3a** with an unexpected loss of sulfonylazide (SO₂N₃) group from TsN₃ (Fig. 2). Optimization of the reaction conditions afforded the yield of **3a** reaching 69% in the presence of 0.5 mol % of [Ir{dt(tBu)₂ppy}₂(dtbbpy)][PF₆] (**PC1**) and 1,4-diazabicyclo (2.2.2) octane (DABCO) (1.5 equiv) in 1,4-dioxane at 80 °C for 24 h under air atmosphere (Fig. 2 Table, entry 1). On the contrary, other catalysts such as fac-Ir(ppy)₃ (**PC2**), Eosin Y, MesAcr⁺ClO₄⁻ (**PC3**), 2,4,6-triphenylpyrylium tetrafluoroborate (TPP), and Ru(bpy)₃Cl₂ (**PC4**) did not give the product **3a** in more than 5% yield (entries 2–6 vs 1). Meanwhile, utilization of different solvent systems (entries 7–9) or inorganic bases such as K₂CO₃ and Cs₂CO₃ (entries 10–11) made this transformation very sluggish. Similarly, running the reaction under room temperature, Ar atmosphere, and green-light irradiation also led to significantly lower yields, respectively (entries 12–14 vs 1). Notably, control experiments under these optimized conditions indicated that 2,3-naphthyl-fused benzosilole **3a** could not be observed at all in the absence of light or photocatalysts (**PC**) (entries 15–16).

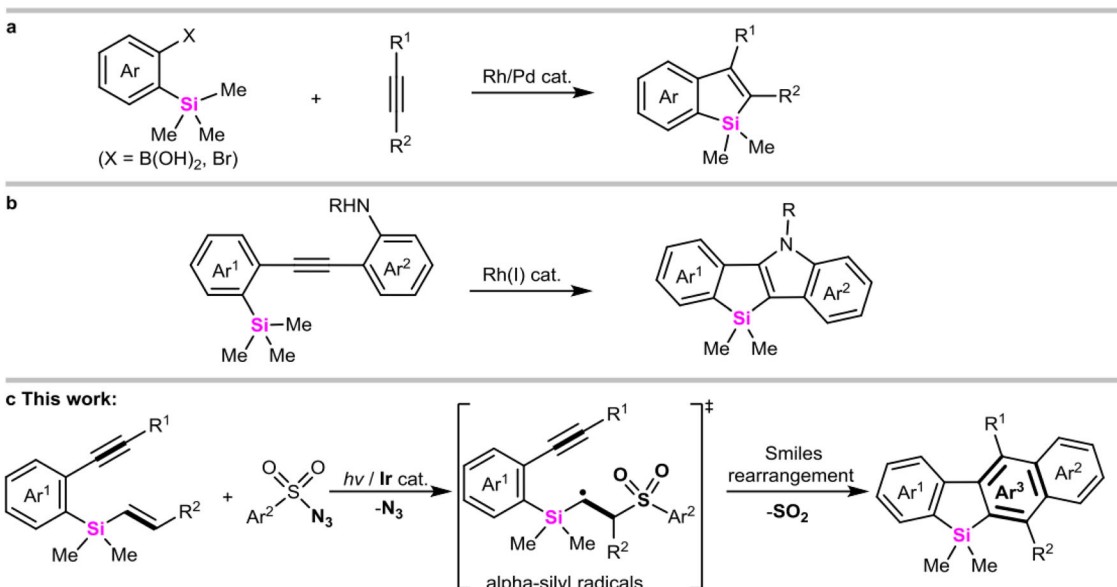

**Fig. 1 Strategies to access benzosiloles. a** Rh/Pd-catalyzed intermolecular cyclovinylation of arylalkylsilanes with alkynes. **b** Rh(I)-catalyzed intramolecular aminosilylation of ortho-alkynylarylsilanes. **c** Photocatalyzed carbocyclization of vinylsilanes with arylsulfonyl azides via alpha-silyl radical Smiles rearrangement.

| Entry | Changes to standard conditions | Yield 3a (%) |
|---|---|---|
| 1 | none | 69 |
| 2 | *fac*-Ir(ppy)$_3$ (**PC2**) | 0 |
| 3 | Eosin Y | < 5 |
| 4 | MesAcr$^+$ClO$_4^-$ (**PC3**) | < 5 |
| 5 | TPP | 0 |
| 6 | Ru(bpy)$_3$Cl$_2$ (**PC4**) | 0 |
| 7 | PhCF$_3$ | 44 |
| 8 | DMF | 0 |
| 9 | CH$_3$CN | trace |
| 10 | K$_2$CO$_3$ | 10 |
| 11 | Cs$_2$CO$_3$ | 15 |
| 12 | room temperature | 38 |
| 13 | Ar atmosphere | 34 |
| 14 | green LEDs | 20 |
| 15 | no light | 0 |
| 16 | no PC | 0 |

**Fig. 2 Reaction development.** All the reactions are conducted in sealed tubes, followed by flash chromatography on SiO$_2$. The isolated yields are reported.

**Substrate scope**. With this optimized protocol, we examined the transformation of various *ortho*-alkynylaryl vinylsilanes (**1**) with *para*-methylphenylsulfonyl azide **2a**. As summarized in Fig. 3, different sila-enynes **1** bearing electron-rich (-Me, -Et, -*n*-Bu, and -MeO), halogen (-F, -Cl and -Br), strong electron-withdrawing (-CF$_3$), and even vinylsilyl substituent at the *para*-position of alkynylphenyl groups (R$^1$ = phenyl), successfully underwent carbocyclization-aromatization with TsN$_3$ (**2a**) to afford 4-(4-substituted-phenyl)-2,3-benzosilafluorenes **3a**, **3b**, **3d**–**3k** in moderate to good yields (50–71%). Likewise, *meta*-methylphenyl-substituted alkynylaryl vinylsilane could react efficiently to provide **3c** in 63% yield. Meanwhile, *ortho*-(1-heptynyl)-phenyl vinylsilane was also tolerable in this reaction to assemble 4-pentyl-substituted-2,3-benzosilafluorenes **3l** (50%). However, when the terminal alkyne-tethered arylvinylsilane was subjected to the reaction system, only a 15% yield of **3m** was obtained. The evaluation of the substituent effect (R$^3$) from vinylsilyl benzene ring of **1** indicated that all of the electron-rich or electron-deficient silylarenes could smoothly react with TsN$_3$ (**2a**) to furnish the desired products **3n**–**3q** with good conversions (58–68% yields), regardless of electronic properties of substituents. It should be noted that hept-1-en-1-yldimethylsilane was not well tolerated under these conditions, only affording 1-pentyl-2,3-benzosilafluorene **3r** in 12% yield.

Interestingly, if vinyldimethylsilane **1** was switched to allyldimethylsilane **4**, an unexpected [2 + 2 + 3] coupling-cyclization of *ortho*-alkynyaryl allyldimethylsilane **4** with arylsulfonyl azides **2** occurred (Fig. 4), affording a novel 6/7/7/6-fused silatetracyclic skeleton in good yields (59–65% for **5a**–**5c**) in which SO$_2$ was kept untouched.

Next, we explored the substrate scope of sulfonyl azides by carrying out the cycloaromatization reaction with vinylsilane **1a** (Fig. 5). Compared with electron-neutral phenylsulfonyl azides (60% for **6a**), 4-MeO, 4-Ph, 4-halo (F, Cl, Br, I), and 4-acetamido-phenylsulfonyl azides could smoothly react with vinylsilane **1a** to produce 2,3-benzosilafluorenes in 21–70% yields (**6d**, **6e**, **6h**, **6i**, **6l**, **6m**, **6n**). Of noted, 3-methylphenylsulfonyl azide, 3-chlorophenylsulfonyl azide, and even 2-naphthylsulfonyl azide furnished 61–74% overall yields of isomers **6b/6c**, **6f/6g**, and **6j/6k**. Gratifyingly, the protocol was still shown to tolerate strong electron-withdrawing (4-CF$_3$, 4-CN, 4-acyl, 4-CO$_2$Et, 4-CO$_2$H) substituted phenylsulfonyl azides and even 3-pyridylsulfonyl azide, giving 42–72% yields of **6o**–**6t**, respectively.

**Application**. To evaluate the potentiality of 2,3-naphthyl-fused benzosiloles in developing organic optoelectronic materials, 8-bromo-5,5-dimethyl-11-phenyl-5H-benzo[b]naphtho[2,3-d]silole **6l** was employed to react with anthracene-9,10-diyldiboronic acid **7** in the presence of Pd-catalysts, and highly π-conjugated ter-anthracene-tethered 1-phenyl-2,3-benzosilafluorene **8** which contains three anthracenyl units, could be obtained in 14% yield

**Fig. 3 Sila-enyne scope.** All the reactions are conducted in sealed tubes, followed by flash chromatography on SiO$_2$. The isolated yields are reported.

**Fig. 4 The coupling-cyclization of allylic dimethylsilanes with arylsulfonyl azides.** All the reactions are conducted in sealed tubes, followed by flash chromatography on SiO$_2$. The isolated yields are reported.

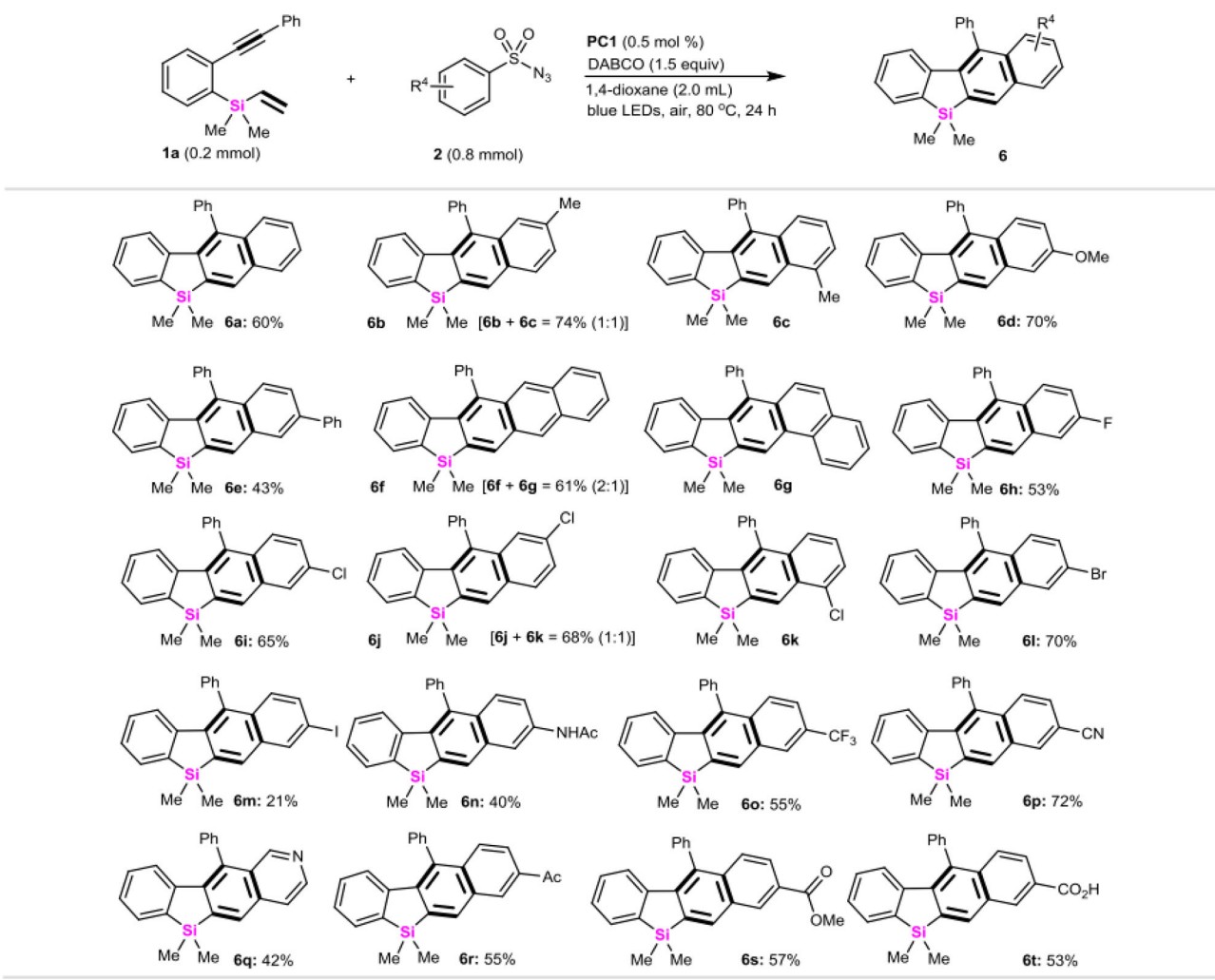

**Fig. 5 Arylsulfonylazide scope.** All the reactions are conducted in sealed tubes, followed by flash chromatography on SiO$_2$. The isolated yields are reported and ratios of isomers are shown in parenthesis.

(Fig. 6a) (it should be noted that this transformation easily led to the protonation of anthracene-9,10-diyldiboronic acid **7**, affording 53% yield of anthracence **9** (see Supplementary Information, pg S26)). Photophysical properties of benzosilole derivative **8** were investigated by ultraviolet-visible (UV-vis) absorption and photoluminescence (PL) spectra in diluted toluene solution ($10^{-5}$ M) (Fig. 6b). The absorption bands in the long-wavelength region (350–400 nm) can be attributed to the π–π* transition of the anthracenyl units[45]. Both PL spectra at r.t. and 77 K exhibited a 0–0 peak at 415 nm, a 0–1 sub-peak at 433 nm, and a 0–2 shoulder at 465 nm. Meanwhile, a PL quantum yield of 51% was achieved, indicating that this molecular skeleton could be a promising building block for deep-blue luminescent materials.

**Mechanistic investigations**. Sulfonyl azides have been rarely converted into sulfonyl radicals in chemical transformation. Up to now, only Konig reported an example that sulfonyl azides were employed as precursors of nitrenes in visible-light photocatalysis[46]. More recently, Lam found that arylsulfonyl azides could be converted into arylsulfonyl radicals under the photocatalytic system, in which hydrogen abstraction from THF was involved[47]. However, our solvent screening indicated that

this cycloaromatization could also be allowed in PhCF$_3$ (Fig. 2, entry 7), in which hydrogen abstraction from PhCF$_3$ is very difficult. The observation implied that sulfonyl azides could possibly produce sulfonyl radical via unstable arylsulfonyldiimide intermediates (ArSO$_2$N=NSO$_2$Ar)[48], which is derived from the nitrene radical dimerization and protonation (this transformation was performed under air conditions, in which the synergistic cooperation of O$_2$, and alkylamines (DABCO), etc. could possibly provide proton sources)[49]. Given that disulfonylhydrazines could produce disulfonyldiimines via single electron transfer (SET) followed by denitrogenation to afford sulfonyl radicals[50,51], we utilized sulfonylhydrazine **10** to react with **1a** (Fig. 7a), the desired product **3a** (30%) could be obtained, this control experiment implied that disulfonyldiimines were possibly involved in this cycloaromatization.

In addition, when 2,2,6,6-tetramethyl-1-piperidinyloxy (TEMPO) (2.0 equiv) was applied to the reaction of vinylsilane **1a** and TsN$_3$, it was found that TEMPO completely inhibited the cycloaromatization of **1a**, and sulfonyl ester **11** (HR-MS: 334.1448, Supplementary Fig. 3) (see Supplementary Information for the HR-MS) could be detected (Fig. 7b), confirming that sulfonyl radicals were involved in this transformation. Interestingly, changing vinylsilane (**1a**) to vinylether (**1u**) produced a seven-membered ring compound **12** (CCDC 2034125) instead of

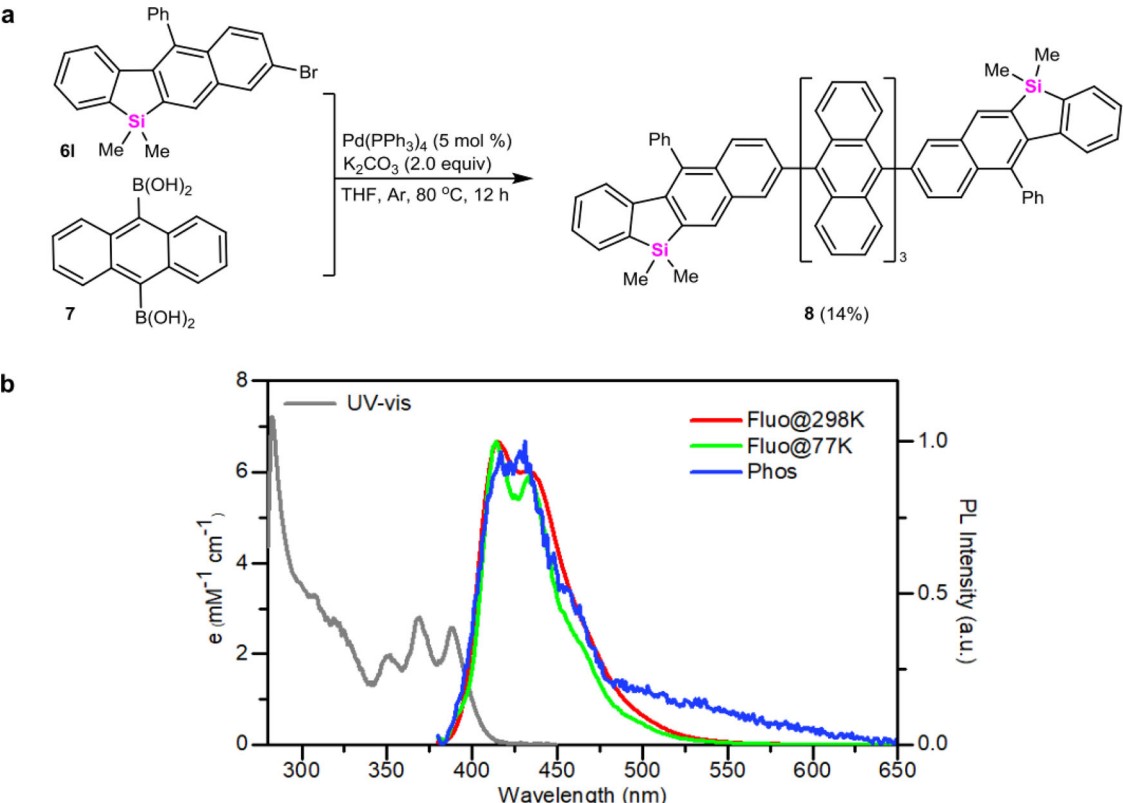

**Fig. 6 Synthetic applications of this method. a** Synthesis of π-conjugated teranthracene-tethered 1-phenyl-2,3-benzosilafluorene **8**. **b** UV-vis absorption and PL spectra of benzosilafluorene **8** measured in diluted toluene solution ($10^{-5}$ M) (a.u. refers to the arbitrary unit. Fluo@298K refers to the fluorescence spectra measured at 298 K. Fluo@77K refers to the fluorescence spectra measured at 77 K. Phos refers to the phosphorescence spectra.).

cycloaromatization product **13** (Fig. 7c). This experiment demonstrated that Si/O switch significantly affected the reaction pathway of vinylsilanes, in which sulfonyl radicals derived from sulfonyl azides preferred to attack carbon-carbon double bonds instead of carbon-carbon triple bonds. Furthermore, compared to *para*- or *meta*-methylphenylsulfonyl azides (**3a** in Fig. 3, **6b**/**6c** in Fig. 5), the coupling-cyclization of *ortho*-methylphe-nylsulfonyl azide **2s** with vinylsilane **1a** only afforded a 16% yield of non-aromatized silacycle **14** (CCDC 2064550), and the aromatized silacycle **6u** was not obtained possibly due to that steric hindrance of *ortho*-methyl substituent of **2s** inhibited a potential structural arrangement and subsequent aromatization (Fig. 7d).

Phosphorescence spectra of the diluted toluene solutions detected at 77 K with a delay time of 0.05 ms were employed to determine the triplet energies of **PC1** (2.58 eV) and TsN₃ (3.10 eV) (Supplementary Fig. 2), demonstrating that an energy transfer (EnT) process between the excited state *Ir(III)-catalyst **PC1** and TsN₃ could not occur. Meanwhile, the reduction potential difference (~0.39 V) between **PC1** ($E_{III/IV}$: −1.45 V versus Ag/AgNO₃ in CH₃CN) and TsN₃ ($E_{red}$: −1.06 V vs Ag/ AgNO₃ in CH₃CN) (see Supplementary Information for the cyclic voltammetry (CV) of TsN₃ and Ir(III)-catalyst **PC1**) means that the electron transfer between them possibly occurred under heating conditions (Supplementary Figs. 4 and 5). Moreover, the fluorescence quenching (Supplementary Figs. 6 and 7) further suggested that a possible SET process between the excited state *Ir(III)-catalyst and TsN₃ led to the formation of sulfonyl radicals via nitrene radicals[52] and arylsulfonyldiimide[48]. Calibrated with DFT calculations, the possible reaction

mechanism is shown in Fig. 8. In Path a, the addition of sulfonyl radical to the carbon-carbon double bond in substrate **1a** results in α-silyl carbon radical **A** (5.6 kcal/mol), which could further undergo the radical cyclization with carbon-carbon triple bond to afford vinylradical **B** (−11.6 kcal/mol). Intramolecular cyclization of **B** at the benzene ring followed by the Smiles rearrangement involved a spirocyclic intermediate **C** (−12.7 kcal/mol) and a subsequent desulfonylation to produce beta-silyl radical **D** (−29.9 kcal/mol). Again, the cascade radical cycliza-tion to **E** (−41.4 kcal/mol), SET, and aromatization give the cycloaromatization product **3a**. In contrast, Ts• radical could attack the carbon-carbon triple bond of **1a** in Path b, leading to ethylenic radical **G** (8.0 kcal/mol), which then forms radical intermediate **H** (−2.0 kcal/mol) and **I** (−4.8 kcal/mol) through cascade cyclization. Although intermediate **I** could be converted to **E** by desulfonylation, control experiments and DFT calcula-tions both suggest Path a to be the more plausible mechanism. Because of that *ortho*-methylphenylsulfonyl azide **2s** was subjected to the standard photocatalyzed system, although *ortho*-methylphenylsulfonyl radical could attack the carbon-carbon double bond of **1a**, the steric hindrance from *ortho*-methyl substituent of **2s** possibly inhibited the formation of the spirocyclic intermediate **C** (Path a), resulting in very poor yield (16%) of non-aromatized silacycle **14** (see Supplementary Information for the HMBC, NOE spectra and single crystal structure (Supplementary Figs. 8, 9, and 15) of compound **14**) through cascade radical cyclization and desulfonylation. This control experiment (Fig. 7d) indirectly implied that the cycloaromatization between vinylsilanes and arylsulfonyl azides via Path b (Fig. 8) was not possible.

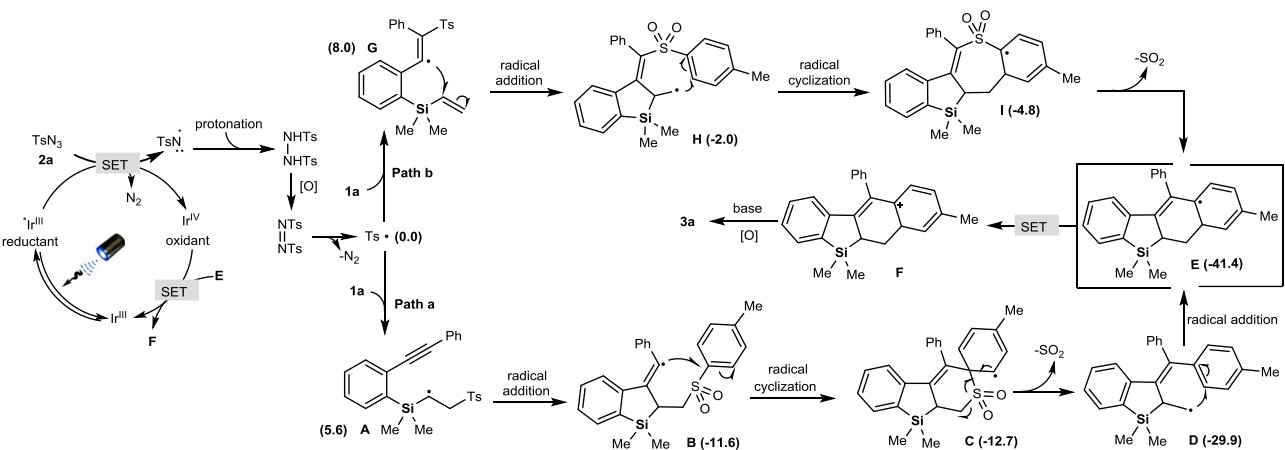

**Fig. 7 Preliminary mechanism studies. a** Photocatalyzed cycloaromatization of vinylsilane **1a** with sulfonylhydrazine **10**. **b** Radical intermediate trapping reaction by TEMPO. **c** Photocatalyzed coupling-cyclization of vinylether **1u** with arylsulfonylazide **2h**. **d** Photocatalyzed coupling-cyclization of vinylsilane **1a** with *ortho*-methylphenylsulfonyl azide **2s**.

**Fig. 8 Proposed reaction mechanism.** Cycloaromatization of vinylsilanes with arylsulfonylazides via Path a and Path b.

## Discussion

In conclusion, we have reported a photoredox-catalyzed cycloaromatization of *ortho*-alkynylaryl vinylsilanes and arylsulfonyl azides, furnishing naphthyl-fused benzosilole skeletons with wide functional group tolerance. This protocol features a unique combination of cascade S-N/C-S bond cleavages and α-silyl radical Smiles rearrangement. These silaarenes show promising potential in luminescent materials, and further application studies of these highly π-conjugated siloles in luminescent materials are undergoing in our lab.

## Methods

**Procedure for the photocatalyzed cycloaromatization of vinylsilanes with arylsulfonylazides.** To a 10 mL vial equipped with a magnetic stir bar, was added vinylsilanes **1** (0.2 mmol), arylsulfonylazides **2** (0.8 mmol), DABCO (33 mg, 0.3 mmol), [Ir{dt(tBu)₂ppy}₂(dtbbpy)][PF₆] **PC1** (1.3 mg, 0.5 mol %) and 1,4-dioxane (2.0 mL) under air conditions. The vial was equipped with a Teflon septum and stirred at 80 °C under blue LED irradiation with two Kessil LEDs (30 W, 456 nm, ~3 cm away from the reaction mixture) for 24 h. The solvent was removed in vacuo and the residue was purified by flash column chromatography on silica gel to yield the desired products.

## Data availability

The authors declare that the data supporting the findings of this study are available within the article as well as from the authors upon reasonable request. The X-ray crystallographic coordinates for structures **3a, 5c, 12**, and **14** reported in this study have been deposited at the Cambridge Crystallographic Data Centre (CCDC) under CCDC 2034071, CCDC 2034073, CCDC 2034125, and CCDC 2064550, respectively. These data can be obtained free of charge from The Cambridge

Crystallographic Data Centre via www.ccdc.cam.ac.uk/data-request/cif. Source data are provided with this paper.

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

## Acknowledgements
We thank NKRDPC (No. 2016YFA0602900), the NSFC (Nos. 21871097, 51625301, 21973113), KARDPGP (No. 2020B010188001), NSFGP (Nos. 2018B030308007, 2019A1515011790), and STPG (No. 201904010113) for financial support.

## Author contributions
W.Z. conceived and directed the project. F.C. performed the experiments. M.L. and C.Y. prepared some substrates. H.J. joined the discussion about this project. Y.S. performed the DFT calculations. Z.K. directed the DFT calculations. S.-J.S. directed the photophysical experiments. W.Z. wrote the manuscript.

## Competing interests
The authors declare no competing interests.
