## [Peer Review File · Nature Communications]

REVIEWER COMMENTS

Reviewer #1 (Remarks to the Author):

This paper describes the synthesis of naphthyl-fused benzosiloles by the iridium-catalyzed, photochemical-induced domino radical sequence between arylsulfonyl azides and ortho-alkynylaryl vinylsilanes. The reactions proceed by the formation of sulfonyl radicals, which add to the vinylsilane to produce an alpha-silyl radical, which then cyclizes onto the alkyne. The resulting alkenyl radical then engages in a radical Smiles rearrangement, and after desulfonylation, a further radical cyclization and oxidation/rearomatization gives the products. A brief evaluation of the potential applications of the products in preparing organic optoelectronic materials was also conducted. The authors have demonstrated the scope of the reaction fairly thoroughly, and have conducted some experiments and DFT calculations to support the proposed reaction mechanism. The chemistry is novel, but is a fairly specialized, complex reaction sequence that produces a specific class of products. On that basis, interest in the actual synthetic method may not be very broad. Nevertheless, the mechanism of the reaction should be of fundamental interest, and the potential utility of benzosiloles in optoelectronics adds some extra value. Therefore, considering the overall balance of results reported, I regard this work as being suitable for publication in Nature Communications. The authors have done a generally good job in producing a solid piece of work (well done).

However, there are a number of points, and potential errors that need to be addressed first:

1. It is arguable whether "silahydrocarbons are often encountered in many pharmaceuticals". Although silicon in bioactive molecules is fairly well-studied, I don't think the proportion of approved pharmaceuticals containing silicon is really that high.
2. This work produces naphthyl-fused benzosiloles. Is there a case for more papers that prepare these products to be cited? E.g. J. Org. Chem. 2019, 84, 957; Adv. Synth. Catal. 2018, 360, 3049; Org. Lett. 2015, 17, 386; Angew. Chem. Int. Ed. 2016, 55, 6319, etc. – and there are various patents as well.
3. In the opinion of this reviewer, the flow of the introduction could be better, in that concepts are presented in a slightly disjointed manner. This refers to the second paragraph on page 2 and the first paragraph of page 3. The authors discuss the general synthetic application of vinylsilanes, aryl migration of Smiles rearrangements, and sulfonyl azides for sulfonyl radical synthesis, which at first may seem to readers to be rather disparate topics. Would it not be better to, after the first paragraph on page 2, discuss the reaction "design" that the authors have developed to prepare naphthyl-fused-benzosiloles and the individual mechanistic steps involved, and then at the appropriate place, related the discussion to broader, literature concepts? That way, readers will be able to see the relevance of those aforementioned topics more easily.
4. Following on from the above point, Scheme 1c may be hard to understand for several reasons:
 - (a) It only shows the structures of the intermediate radicals, and not of the corresponding products.
 - (b) It appears before Scheme 1d, so people just scanning the paper and looking at the Scheme may wonder about the relevance of Scheme 1c.
 - (c) It is not consistent with the Scheme 1 legend – Scheme 1c is not about making benzosiloles. I think the authors should think about whether Scheme 1c should be moved to a different place (later), or is actually needed at all. If it is decided that the diagram should stay, then more detail may be needed to be provided, depending on how the authors have decided to change things.
5. In Table 1 and the accompanying text, the authors should not use the abbreviations PC1, PC2, etc. Just write out the photocatalyst in full. It is so much easier to understand and doesn't really add much space.
6. The text and Table 1 doesn't mention any details of the light source used. From the later Schemes, it appears that blue LEDs were used. This should be mentioned in the text and shown in Table 1.
7. The compound numbering system could be simpler. For example, the use of numbers like "3-1a" could confuse because the dash is commonly taken to mean that a range of different compounds are being referred to. And this is why the authors have had to use the symbol ~

instead of a normal dash, to avoid confusion. Perhaps the authors can adopt a simpler system?
8. There are a number of instances where "trimethylsilyl" has been used, when it should actually be "dimethylsilyl".

9. For the three products derived from an allylic dimethylarylsilane (products 3-1s, 3-1t, and 3-1u), these results should be written out in a separate Scheme that shows the structures of the starting materials, and perhaps the reaction mechanism can be included somewhere. These should not be included in Scheme 2. They use completely different starting materials and give different products to that shown in the generic Scheme 2 reaction at the top.

10. For greater clarity/conciseness, the equivalents of the sulfonyl azide should be shown in the reaction diagrams, and the Scheme footnotes can be shortened to remove information that has already been shown in the picture (e.g. PC catalyst loading and DABCO equivalents, reaction temperature and time). There is no need to duplicate information.

11. When discussing the formation of product 5, the authors simply say it was "unexpected". There should be an explicit statement that the product contains three units derived from the bis-boronic acid. What was the remainder of the material in this reaction? Are there products derived from the incorporation of one or two equivalents of the bis-boronic acid?

12. In Scheme 5, product 10 is obtained from the reaction of ortho-methylphenylsulfonyl azide. However, when the authors are discussing the mechanism of the general reaction on pages 9 and 10, referring to Scheme 6, they mention that product 10 is formed by following path b of the reaction mechanism. This appears totally incorrect, because compound 10, as drawn, does not have the same structure/connectivity as intermediate I in Scheme 6. The authors should double-check all of this, including whether they have assigned the structure of compound 10 correctly (is there unambiguous structural evidence)? If the structure of compound 10 is correct, then the authors need to change their explanation, and the mechanism followed to give compound 10 should preferably be included somewhere. (I think that the structure of compound 10 is probably correct.) Also, the authors mention compound 9 is produced in 13% yield, when they actually mean compound 10.

13. In the SI, it would be nice to mention what solvent systems were used to purify the products during column chromatography, as well as stating how much of each starting material (in mg) was used in each reaction.

Reviewer #2 (Remarks to the Author):

The authors report in this manuscript an unprecedented photoredox-catalyzed cycloaromatization of ortho-alkynylaryl vinylsilanes with arylsulfonyl azides. A large number of 2,3-naphthyl-fused benzosiloles were obtained in moderate to high yields by varying systematically the nature and positions of substituents in both reaction partners. A plausible reaction mechanism was proposed based on the results of the control experiments and the DFT calculations. The protocol featuring a cascade S-N/C-S bond cleavages/radical Smiles rearrangement is unprecedented. Interestingly, switching vinyltrimethylsilanes to allylic trimethylsilanes, the reaction afforded the compounds with a bridged silatetracyclic compounds (3-1s ~ 3-1u) via an apparently different reaction pathway. It is an interesting approach to benzosiloles and I recommend its publication in *Nat. Commun.* after minor revision noted below:

Page 3, lines 4-5: "Recently, it was found that sulfonylazides are potential nitrogen radical precursors to enable aryl C-H amination⁴⁰ under photocatalysis systems" It is clearly stated in the cited paper (ref 40) that the radical is not involved in the C-H amination reaction ("Addition of TEMPO does not interfere with the reaction, which also indicates the absence of radical intermediates in this reaction."). Therefore, this statement as well as the citation is irrelevant.

Page 9, 2nd paragraph, lines 2-3: "the fluorescence quenching (Fig. S3) suggested a possible energy transfer (EnT) process between the excited state *Ir(III)-catalyst and TsN3" It would be nice if authors could give the triplet energy of PC1 and TsN3 to support their assumption.

For products 3-2b/3-2c, 3-2f/3-2g, 3-2j/3-2k in Scheme 3: is it possible to separate the two isomers? If not, the ratio of the two isomers should be given in Scheme 3.

Typo errors:

Page 4, last line: change "alknylphenyl group" to "alkynylphenyl group"

Page 5, line 3: change "ortho-(1-heptynyl)-phenyl vinylsilane" to "ortho-(1-heptynyl)-phenyl vinylsilane".

Page 5, line 8: change "It should noted that" to "It should be noted that".

Annotation for Scheme 2: "using vinylsilane 1a (0.20 mmol)", change "1a" to "1".

Scheme 3, footnote: change "using vinylsilane 1a (0.20 mmol), TsN3 (0.40 mmol)", change "TsN3* to 2 (0.40 mmol)".

Page 8: "disulfonylhydrazines could produce sulfonyldiimines" and "this control experiment suggested that sulfonyldiimines": change "sulfonyldiimines" to "disulfonyldiimides".

Reference:

Ref 40 and 42 are the same.

Ref 32: change "Org. Chem. 82, 4449 – 4457 (2017)" to "J. Org. Chem. 82, 4449 – 4457 (2017) "

SI

1g: 162.5 (d, J = 48 Hz, 2JCF), 133.1 (d, J = 11 Hz, 2JCF), 119.6 (d, J = 3 Hz, 3JCF), 115.8 (d, J = 22 Hz, 2JCF) are wrong, please double check.

1j: 131.8 (d, J = 6 Hz, 3JCF), 129.9 (d, J = 130 Hz, 2JCF), 129.9 (d, J = 33 Hz, JCF), 128.1 (d, J = 198 Hz, 1JCF), 125.4 (q, J = 8 Hz, JCF): they should be quartets. Please check also the coupling constants.

1o, 1q: check the ¹³C NMR. Please double check the C-F coupling constants.

Please double check the ¹³C NMR of all of the other fluorine-containing compounds: 3-1g, 3-1j, 3-1o, 3-1q, 3-1u, 3-2h, 3-2o, paying particular attention on the splitting pattern and the coupling constants.

The Responses to the Reviewers Comments

Journal: *Nat. Commun.*

Title: *Photocatalyzed Cycloaromatization of Vinylsilanes with Arylsulfonylazides*

Author(s): Fengjuan Chen,^{a,§} Youxiang Shao,^{b,§} Mengke Li,^{c,§} Can Yang,^a Shi-Jian Su,^{*c} Huanfeng Jiang,^a Zhuofeng Ke,^{*b} and Wei Zeng^{*a}

Address:

^a Key Laboratory of Functional Molecular Engineering of Guangdong Province, School of Chemistry and Chemical Engineering, South China University of Technology, Guangzhou 510641, China

^b School of Materials Science and Engineering, PFCM Lab, Sun Yat-sen University, Guangzhou 510275, China

^c State Key Laboratory of Luminescent Materials and Devices, Institute of Polymer Optoelectronic Materials and Devices, South China University of Technology, Guangzhou 510641, China

Dear Reviewers:

The above-mentioned manuscript was ever submitted to *Nat. Commun.* for peer review, and we got the corresponding review results on Jan. 5, 2021. Now this manuscript has already been revised according to your kind suggestions. All of the revisions and updates are shown as follows.

A. Respond to the 1st reviewer's comments

Question 1: It is arguable whether “silahydrocarbons are often encountered in many pharmaceuticals”. Although silicon in bioactive molecules is fairly well-studied, I don't think the proportion of approved pharmaceuticals containing silicon is really that high.

Respond to this question: Thanks the 1st reviewer's suggestion, we already changed this sentence “silahydrocarbons are often encountered in many pharmaceuticals...” into “silahydrocarbons are sometimes encountered in pharmaceuticals ...”, please check it in our revised manuscript.

Question 2: This work produces naphthyl-fused benzosiloles. Is there a case for more papers that prepare these products to be cited? E.g. *J. Org. Chem.* 2019, 84, 957; *Adv. Synth. Catal.* 2018, 360, 3049; *Org. Lett.* 2015, 17, 386; *Angew. Chem. Int. Ed.* 2016, 55, 6319, etc. - and there are various patents as well.

Respond to this question: We already cited these papers in ref. 19-22, please check our revised manuscript, thanks.

Question 3: Would it not be better to, after the first paragraph on page 2, discuss the reaction “design” that the authors have developed to prepare naphthyl-fused-benzosiloles and the individual mechanistic steps involved, and then at the appropriate place, related the discussion to broader, literature concepts? That way, readers will be able to see the relevance of those aforementioned topics more easily.

Respond to this question: We already reorganized the 2nd paragraph on Pg 2 and the 1st paragraph of Pg 3 by combining “the introduction of vinylsilanes” with “the reaction design” as follows, please check it, thanks.

“...photophysical properties. Thus, there has been an ever-increasing demand for the rapid assembly of diversified polycycle-fused siloles.^{17,18, 19, 20,21, 22}

Aryl migration *via* Smiles rearrangement is a powerful tool for the synthesis of polycyclic arenes.²³⁻²⁵ However, the modes of radical Smiles rearrangement are very limited. Up to now, only α -carbonyl radical,²⁶⁻³⁰ β -aminoalkyl radical,³¹⁻³³ *N*-centered radical,³⁴ and ketyl radical³⁵-triggered Smiles rearrangement have been exploited to construct nitrogen-heterocycles, and vinylsilane-based Smiles rearrangement keep unexplored. As is well known, vinylsilanes have proven to be important “alkene” sources in Hiyama coupling, which could efficiently incorporate C=C bond in a particular molecule with the release of silyl moiety;³⁶ Meanwhile, the high electronegativity of carbon (2.35) relative to silicon (1.64)³⁷ and silyl hyperconjugation effect (the so-called β -effect)³⁸⁻⁴⁰ generally endow these compounds with the versatile reactivity. For examples, Jun⁴¹ reported that Rh(I)-catalyzed cross-coupling of aldehydes with vinylsilanes led to the formation of β -acylsilanes *via* β -silylethylrhodium(III) intermediates. On the contrary, Buchwald⁴² and Miura⁴³ demonstrated that Cu(I)-catalyzed addition-coupling of vinylsilanes with amines could produce α -aminosilanes. Thus it can be seen that developing new vinylsilane-based coupling-cyclization will possibly establish a new platform to assemble complex silacycles. Again, azides could be employed as potential nitrogen radical precursors to enable C-H amination⁴⁴ under photocatalysis systems. Accordingly, visible-light catalyzed coupling of vinylsilanes with arylsulfonylazides could possibly generate α -silyl radicals and initiate the silylalkene Smiles rearrangement. To explore this feasibility, herein we describe an unprecedented cycloaromatization of *ortho*-alkynylarylsilylalkenes with arylsulfonyl azides for rapid assembly of 2,3-naphthyl-fused benzosiloles *via* a cascade S-N/C-S bond cleavage in the presence of visible-light (**Scheme 1c**).”

Question 4: Following on from the above point, Scheme 1c may be hard to understand for several reasons:

- It only shows the structures of the intermediate radicals, and not of the corresponding products.
- It appears before Scheme 1d, so people just scanning the paper and looking at the Scheme may wonder about the relevance of Scheme 1c.
- It is not consistent with the Scheme 1 legend - Scheme 1c is not about making benzosiloles.

I think the authors should think about whether Scheme 1c should be moved to a different place (later), or is actually needed at all. If it is decided that the diagram should stay, then more detail may be need to be provided, depending on how the authors have decided to change things.

Respond to this question: Thanks for the referee’s good suggestions, we thought we had better move away the Scheme 1c from our revised manuscript as follows, and just cited the corresponding references to make the Scheme 1 more concise, please check it, thanks.

Scheme 1. Strategies to Access Benzosiloles

Question 5: In Table 1 and the accompanying text, the authors should not use the abbreviations PC1, PC2, etc. Just write out the photocatalyst in full. It is so much easier to understand and doesn't really add much space.

Respond to this question: In Table 1, we provided the exact structures about **PC1-PC4**, and wrote out the photocatalyst in full in our revised manuscript, please check it, thanks.

Question 6: The text and Table 1 doesn't mention any details of the light source used. From the later Schemes, it appears that blue LEDs were used. This should be mentioned in the text and shown in Table 1.

Respond to this question: In Table 1, we provided the exact light source (blue LEDs) in our revised manuscript, please check it, thanks.

Question 7: The compound numbering system could be simpler. For example, the use of numbers like "3-1a" could confuse because the dash is commonly taken to mean that a range of different compounds are being referred to. And this is why the authors have had to use the symbol ~ instead of a normal dash, to avoid confusion. Perhaps the authors can adopt a simpler system.

Respond to this question: To make the compound numbering system simpler, we used "3" to replace "3-1" (Table 1 and Scheme 2), and used "6" to replace "3-2" (Scheme 4) to number the products in our revised manuscript, please check it, thanks.

Question 8: There are a number of instances where "trimethylsilyl" has been used, when it should actually be "dimethylsilyl".

Respond to this question: We already changed "trimethylsilyl" into "dimethylsilyl" in our revised manuscript, please check it, thanks.

Question 9: For the three products derived from an allylic dimethylarylsilane (products 3-1s, 3-1t, and 3-1u), these results should be written out in a separate Scheme that shows the structures of the starting materials, and perhaps the reaction mechanism can be included somewhere. These should not be included in Scheme 2. They use completely different starting materials and give different products to that shown in the generic Scheme 2 reaction at the top.

Respond to this question: We already separated **Scheme 2** into **Scheme 2** and **Scheme 3** in our revised manuscript, and the **Scheme 3** only contained the three products derived from an allyldimethylarylsilanes **5a-5c**. Moreover, we also proposed a possible reaction mechanism in SI (**Fig. S10**, Pg S34), please check it, thanks.

Question 10: For greater clarity/conciseness, the equivalents of the sulfonyl azide should be shown in the reaction diagrams, and the Scheme footnotes a can be shortened to remove information that has already been shown in the picture (e. g. PC catalyst loading and DABCO equivalents, reaction temperature and time). There is no need to duplicate information.

Respond to this question: According to the referee's suggestions, we already reorganized all the Schemes in our revised manuscript to make the Scheme footnotes more concise, please check them, thanks.

Question 11: When discussing the formation of product 5, the authors simply say it was "unexpected". There should be an explicit statement that the product contains three units derived from the bis-boronic acid. What was the remainder of the material in this reaction? Are there products derived from the incorporation of one or two equivalents of the bis-boronic acid?

Respond to this question: We already changed this sentence to "..., and highly π -conjugated teranthracene-tethered 1-phenyl-2,3-benzosilafluorene **8** which contains three anthracenyl units, could be obtained in 14% yield (**Scheme 5a**).⁴⁵...".

Moreover, the remainder of the material in this reaction belonged to **anthracene 9** which derived from the protonation of **anthracene-9,10-diyldiboronic acid 7**. We already noted this case in ref. 45, please check it, thanks.

In addition, we repeated this reaction again, and no one anthracenyl unit-containing or two anthracenyl units-containing benzosilole derivatives were observed.

Question 12: In Scheme 5, product 10 is obtained from the reaction of ortho-methylphenylsulfonyl azide. However, when the authors are discussing the mechanism of the general reaction on pages 9 and 10, referring to Scheme 6, they mention that product 10 is formed by following path b of the reaction mechanism. This appears totally incorrect, because compound 10, as drawn, does not have the same structure/connectivity as intermediate I in Scheme 6. The authors should double-check all of this, including whether they have assigned the structure of compound 10 correctly (is there unambiguous structural evidence)? If the structure of compound 10 is correct, then the authors needs to change their explanation, and the mechanism followed to give compound 10 should preferably be included somewhere. (I think that the structure of compound 10 is probably correct.) Also, the authors mention compound 9 is produced in 13% yield, when they actually mean compound 10.

Respond to this question: Thanks for the referee's good suggestions. After we tested the HMBC spectra and NOE spectra of compound **14** (see **Fig. S8** and **Fig. S9** in Pg S33-PgS34), and found that our previous structure of compound **10** was wrong; the exact structure of this compound **10** should be compound **14**, please check it in our revised manuscript.

The previous structure of compound **10**

The corrected structure of compound **14**

Moreover, to make the discussion part about the formation of compound **14** simpler, we revised this part as follows:

“... Because of that *ortho*-methylphenylsulfonyl azide was subjected to the standard photocatalyzed system, *ortho*-methyl substituent of **2s** inhibited the formation of the spirocyclic intermediate **C** (**Path a**) due to steric hindrance. Therefore, *ortho*-methylphenylsulfonyl radical could only attack the carbon-carbon triple bond of **1a**, leading to the formation of a seven-membered silacycle **14**⁵⁶ with poor yield (13%) through cascade radical cyclization and aromatization process (**Scheme 6d**), in which desulfonylation did not occur (**Path b**). This control experiment further...”

Question 13: In the SI, it would be nice to mention what solvent systems were used to purify the products during column chromatography, as well as stating how much of each starting material (in mg) was used in each reaction.

Respond to this question: In the SI, we noted the solvent systems which were used to purify the products during column chromatography; and also stated how much of each starting material (in mg) was used in each reaction, please check our revised manuscript, thanks.

B. Respond to the 2nd reviewer's comments

Question 1: lines 4–5: “Recently, it was found that sulfonylazides are potential nitrogen radical precursors to enable aryl C–H amination⁴⁰ under photocatalysis systems” It is clearly stated in the cited paper (ref 40) that the radical is not involved in the C–H amination reaction (“Addition of TEMPO does not interfere with the reaction, which also indicates the absence of radical intermediates in this reaction.”). Therefore, this statement as well as the citation is irrelevant.

Respond to this question: Thanks the 2nd reviewer's suggestion, we already cited another work about photocatalyzed C–H amination via nitrogen radicals, which were derived from azides (see Ref. 44, *J. Am. Chem. Soc.* **2016**, *138*, 12636 – 12642), please check it in our revised manuscript.

Question 2: Page 9, 2nd paragraph, lines 2–3: “the fluorescence quenching (Fig. S3) suggested a possible energy transfer (EnT) process between the excited state *Ir(III)-catalyst and TsN3” It would be nice if authors could give the triplet energy of PC1 and TsN3 to support their assumption.

Respond to this question: Thanks the 2nd reviewer for his good suggestion, we tested the triplet energy of **PC1** (2.58 eV) and TsN₃ (2.97 eV) (see **Fig. S2**), and found that a energy transfer (EnT) process between the excited state *Ir(III)-catalyst **PC1** and TsN₃ could not occur. Again, we carefully tested the reduction potential of **PC1** ($E_{III/IV}$: -1.45 V versus Ag/AgNO₃ in CH₃CN) and TsN₃ (E_{red} : -1.06 V versus Ag/AgNO₃ in CH₃CN) by using Ferrocene as internal standard (see **Fig. S4** and **Fig. S5**), the reduction potential difference (~0.39 V) between **PC1** ($E_{III/IV}$: -1.45 V versus Ag/AgNO₃ in CH₃CN) and TsN₃ (E_{red} : -1.06 V versus Ag/AgNO₃ in CH₃CN)⁵⁴ means that the electron transfer between them possibly occurred under heating conditions. Moreover, the fluorescence quenching (**Fig. S6** and **Fig. S7**) further suggested that a possible single electron transfer (SET) process between the excited state *Ir(III)-catalyst and TsN₃ led to the formation of sulfonyl radicals via nitrene radicals (*Chem. Asian J.* **2018**, *13*, 255-260) and arylsulfonyldiimide (*Tetrahedron Lett.* **1995**, *36*, 1221 – 1222). Finally, we re-proposed a possible reaction mechanism about the formation of sulfonyl radicals in **Scheme 7**, please check it in our revised manuscript.

Question 3: For products 3-2b/3-2c, 3-2f/3-2g, 3-2j/3-2k in Scheme 3: is it possible to separate the two isomers? If not, the ratio of the two isomers should be given in Scheme 3.

Respond to this question: The isomers of **3-2b/3-2c**, **3-2f/3-2g**, **3-2j/3-2k** in **Scheme 4** could not be separated to obtain pure products, we already provided the corresponding ratio of two isomers in Scheme 4, please check them, thanks.

Question 4: Typo errors:

Page 4, last line: change “alknylphenyl group” to “alkynylphenyl group”

Page 5, line 3: change “ortho-(1-heptynyl)-phenyl vinylsilane” to “ortho-(1-heptynyl)-phenyl vinylsilane”.

Page 5, line 8: change “It should noted that” to “It should be noted that”.

Annotation for Scheme 2: “using vinylsilane 1a (0.20 mmol)”, change “1a” to “1”.

Scheme 3, footnote: change “using vinylsilane 1a (0.20 mmol), TsN₃ (0.40 mmol)”, change “TsN₃* to 2 (0.40 mmol)”.

Page 8: “disulfonylhydrazines could produce sulfonyldiimines” and “this control experiment suggested that sulfonyldiimines”: change “sulfonyldiimines” to “disulfonyldiimides”.

Respond to this question: We already corrected the above-mentioned errors, please check them, thanks.

Question 5: Reference: Ref 40 and 42 are the same.

Ref 32: change “Org. Chem. 82, 4449 – 4457 (2017)” to “J. Org. Chem. 82, 4449 – 4457 (2017)”

Respond to this question: We already changed re 42 to 44 (Ref. 44. Huang, X.; Webster, R. D.; Harms, K.; Meggers, E. Asymmetric catalysis with organic azides and diazo compounds initiated by photoinduced electron transfer. *J. Am. Chem. Soc.* **138**, 12636 – 12642 (2016); changed ref 32 to Ref. 28 (Huang, H. & Li, Y.-J. Sustainable difluoroalkylation cyclization cascades of 1,8-enynes. *J. Org. Chem.* **82**, 4449 – 4457 (2017), please check them, thanks.

Question 6: SI: 1g: 162.5 (d, J = 48 Hz, 2JCF), 133.1 (d, J = 11 Hz, 2JCF), 119.6 (d,

J = 3 Hz, 3JCF), 115.8 (d, J = 22 Hz, 2JCF) are wrong, please double check.
1j: 131.8 (d, J = 6 Hz, 3JCF), 129.9 (d, J = 130 Hz, 2JCF), 129.9 (d, J = 33 Hz, JCF), 128.1 (d, J = 198 Hz, 1JCF), 125.4 (q, J = 8 Hz, JCF): they should be quartets. Please check also the coupling constants.
1o, 1q: check the ^{13}C NMR. Please double check the C-F coupling constants.
Please double check the ^{13}C NMR of all of the other fluorine-containing compounds: 3-1g, 3-1j, 3-1o, 3-1q, 3-1u, 3-2h, 3-2o, paying particular attention on the splitting pattern and the coupling constants.

Respond to this question: We already carefully checked all the ^{13}C NMR about F-containing compounds, and provided the corrected data of splitting pattern and the coupling constants, please check our revised manuscript, thanks very much again.

Finally, we also provided the revised versions with changes highlighted as “The Related Manuscript for The Editors Only”, we would like to submit this work to *Nat. Commun.* for peer review, and hope this manuscript can be published in *Nat. Commun.* If any questions please let me know, thanks.

Yours Sincerely

Prof. Wei Zeng
School of Chemistry and Chemical Engineering
South China University of Technology
Guangzhou 510641, P. R. China

REVIEWER COMMENTS

Reviewer #1 (Remarks to the Author):

In this revised manuscript, the authors have made a comprehensive attempt to address the points highlighted by the original reviewers. After examining the changes made to the paper and Supporting Information, this reviewer is satisfied that most of the points addressed to a high standard. The only remaining points that I would like to comment upon/question are as follows:

1. The Table footnotes "a" should be shortened even further to state that the reactions were conducted in sealed tubes and (optionally) to say that products were purified by flash chromatography on SiO₂. It is not necessary to restate the reagents used in the reaction because this information is in the diagrams already. There's no need to duplicate it.
2. Regarding compound 14 in the revised manuscript (compound 10 in the original manuscript), this reviewer asked for the authors to double-check the structure of the compound, but I also said that I thought the original structure 10 is probably correct. Chemically, this makes more sense to me, as I can't see why adding an ortho-methyl group to the arylsulfonyl radical would switch the reaction pathway from attacking a terminal alkene carbon (which is sterically unhindered) to attack the diarylalkyne, which, to me, seems more hindered. The authors have obtained HMBC spectra to support the assignment of the revised structure 14, but I would just like to ensure that the authors are 100% sure about this assignment. The correlations are complicated and I did not take the time to go through all of this myself, so the authors should recheck this. Is the compound crystalline and can an X-ray structure be obtained? Personally, I think it would make more chemical sense if this product had the structure of compound 10 in the original paper. The reaction pathway to form this would be initially through pathway a. But because of the steric effect of the ortho-methyl group, the Smiles rearrangement by attack of the ipso-carbon is disfavored, and instead the alkenyl radical attacks the other (unsubstituted) ortho-carbon, leading to compound 10.

Reviewer #2 (Remarks to the Author):

The authors have satisfactorily addressed the experimental and other items requested by the reviewers. In particular, the careful re-examination of the reaction mechanism (the initiation step) is appreciated. My recommendation is publication of the article with no further revisions.

The Responses to the Reviewer Comments

Journal: *Nat. Commun.*

Manuscript ID: NCOMMS-20-47593A

Title: *Photocatalyzed Cycloaromatization of Vinylsilanes with Arylsulfonylazides*

Author(s): Fengjuan Chen,^{a,§} Youxiang Shao,^{b,§} Mengke Li,^{c,§} Can Yang,^a Shi-Jian Su,^{*,c} Huanfeng Jiang,^a Zhuofeng Ke,^{*,b} and Wei Zeng^{*,a}

Address:

^a Key Laboratory of Functional Molecular Engineering of Guangdong Province, School of Chemistry and Chemical Engineering, South China University of Technology, Guangzhou 510641, China

^b School of Materials Science and Engineering, PFCM Lab, Sun Yat-sen University, Guangzhou 510275, China

^c State Key Laboratory of Luminescent Materials and Devices, Institute of Polymer Optoelectronic Materials and Devices, South China University of Technology, Guangzhou 510641, China

Dear the 1st Reviewer:

The above-revised manuscript was ever submitted to *Nat. Commun.* for peer review, and we got the corresponding review results on Feb. 8, 2021. Now this manuscript has already been revised according to your kind advices. All of the revisions and updates are shown as follows.

A. Respond to the 1st Reviewer's Comments

Question 1: The Table footnotes "a" should be shortened even further to state that the reactions were conducted in sealed tubes and (optionally) to say that products were purified by flash chromatography on SiO₂. It is not necessary to restate the reagents used in the reaction because this information is in the diagrams already. There's no need to duplicate it.

Respond to this question: We already corrected all the Table footnotes "a" as follows, please check it.

^aAll the reactions were conducted in sealed tubes, followed by flash chromatography on SiO₂.

Question 2: The authors have obtained HMBC spectra to support the assignment of the revised structure **14**, ... Is the compound crystalline and can an X-ray structure be obtained?

Respond to this question: We already got the corresponding single crystal data of the compound **14**. Thanks very much for the 1st reviewer's suggestion, he pushed us to confirm the exact chemical structure of compound **14** by the cultivation of single crystal. For the compound **14** in **Scheme 1a**, the ¹H NMR (see SI), ¹³C NMR (see SI), HMBC (see SI) and NOE (see SI) spectrum look reasonable, and the fragment ion at m/z 417.2167 in HR-MS (**Fig. 1d**) misled us to think that the structure of the compound was **14** in **Scheme 1a**. In fact, if the structure of the product belongs to the compound **14** in **Scheme 1b**, the ¹H NMR (see SI), ¹³C NMR (see SI), HMBC (see SI) and NOE (see SI) also look reasonable. But the corresponding single crystal data of compound **14** finally confirmed that the structure of **14** in **Scheme 1b** is right (see **Fig.2**), the structure of **14** in **Scheme 1a** is wrong. Meanwhile, after we carefully check the HR-MS spectrum of the compound **14**, and found that the fragment ion at m/z 353.1715 in HR-MS (see **Fig. 1b**) was just the molecular ion peak of the compound **14** in **Scheme 1b**.

B40_20200903091418 #22 RT: 0.22 AV: 1 NL: 7.47E7
T: FTMS - c APCI corona Full ms [100.0000-1500.0000]

a

B40_20200903091418 #21 RT: 0.21 AV: 1 NL: 4.04E9
T: FTMS + c APCI corona Full ms [100.0000-1500.0000]

b

B40_20200903091418 #21 RT: 0.21 AV: 1 NL: 1.82E8
T: FTMS + c APCI corona Full ms [100.0000-1500.0000]

c

Fig. 1 HR-MS spectrum of compound **14**

By the way, for the possible formation mechanism of the product **14** in Scheme 3, we agree with the 1st reviewer's opinion. If this reaction (**Scheme 3b**) proceeded *via* **Path b**, then all the products **3a-3r** and **6a-6t** in the main text should be the non-aromatized silacycles, it is not possible that *ortho*-methyl substituent of **2s** just changed the reaction pathway. Meanwhile, the yield of **14** should not be very poor. On the contrary, all the products **3a-3r** and **6a-6t** in our manuscript are aromatized products, so the formation of product **14** should not proceed *via* **Path b**. Therefore, based on the control experiments and our DFT calculations in our main text, we think that this reaction (**Scheme 3b**) proceeded *via* **Path a**, and the steric hindrance from *ortho*-methyl substituent of **2s** possibly inhibited and delayed the formation of the spirocyclic intermediate **C** (**Path a**), resulting in very poor yield (16%) of non-aromatized silacycle **14**.

The possible reaction pathway of compound 14 (Path b):

The possible reaction pathway of compound 14 (Path a):

Scheme 3. The possible reaction mechanism for the product 14

Finally, we re-calculated the yield of compound 14 (16%) based on the molecular weight of compound 14, and revised our manuscript to some degree according to the single crystal data of compound 14, please check it.

Thanks the 1st reviewer very much once again.

Yours Sincerely

Prof. Wei Zeng
 School of Chemistry and Chemical Engineering
 South China University of Technology
 Guangzhou 510641, P. R. China

REVIEWERS' COMMENTS

Reviewer #1 (Remarks to the Author):

The authors have satisfactorily addressed the comments and it is pleasing that the correct structure of compound 14 was finally obtained using X-ray crystallography, supported by other data.

The manuscript can be published now, and it will make a good paper in Nature Communications.

Just a final comment - in the response to reviewers, in Scheme 3, it is not correct to display different structures C and C' with the methyl group having a clash with either the Ph group or the sulfone. Because the spirocyclic center is tetrahedral (sp^3 -hybridised carbon), the methyl group will either be above or below the plane of the cyclic sulfone ring, and it will be approximately equal in distance between the Ph and SO₂ groups. If the stereochemistry at the spirocyclic center is drawn out and considered, this will be clearer. In any case, this does not really matter as I assume Scheme 3 in the response to reviewers does not appear in manuscript or the SI (at least I could not find out).

The Responses to the Reviewer Comments

Journal: *Nat. Commun.*

Manuscript ID: NCOMMS-20-47593B

Title: *Photocatalyzed Cycloaromatization of Vinylsilanes with Arylsulfonylazides*

Author(s): Fengjuan Chen^{1,§}, Youxiang Shao^{2,§}, Mengke Li^{3,§}, Can Yang¹, Shi-Jian Su^{3*}, Huanfeng Jiang¹, Zhuofeng Ke^{2*}, and Wei Zeng^{1*}

Address:

¹ Key Laboratory of Functional Molecular Engineering of Guangdong Province, School of Chemistry and Chemical Engineering, South China University of Technology, Guangzhou 510641, China

² School of Materials Science and Engineering, PFCM Lab, Sun Yat-sen University, Guangzhou 510275, China

³ State Key Laboratory of Luminescent Materials and Devices, Institute of Polymer Optoelectronic Materials and Devices, South China University of Technology, Guangzhou 510641, China

Dear the 1st Reviewer:

The above-revised manuscript was ever submitted to *Nat. Commun.* for peer review, and we got the corresponding review results on Mar. 4, 2021. Thank you very much for your constructive suggestions to this manuscript..

A. Respond to the 1st Reviewer's Comments

Question 1: Reviewer #1 (Remarks to the Author):

The authors have satisfactorily addressed the comments and it is pleasing that the correct structure of compound 14 was finally obtained using X-ray crystallography, supported by other data.

The manuscript can be published now, and it will make a good paper in Nature Communications.

Just a final comment – in the response to reviewers, in Scheme 3, it is not correct to display different structures C and C' with the methyl group having a clash with either the Ph group or the sulfone. Because the spirocyclic center is tetrahedral (sp³-hybridised carbon), the methyl group will either be above or below the plane of the cyclic sulfone ring, and it will be approximately equal in distance between the Ph and SO₂ groups. If the stereochemistry at the spirocyclic center is drawn out and considered, this will be clearer. In any case, this does not really matter as I assume Scheme 3 in the response to reviewers does not appear in manuscript or the SI (at least I could not find out).

Respond to this question: For the previous response to reviewers, in Scheme 3, I agree with the 1st reviewer's opinion about the structures C and C' with the methyl group having a similar clash with either the Ph group or the sulfone. Considering that this part will not appear in manuscript or the SI, so we just stated that we agree with and understand the 1st reviewer's opinion. Thanks the 1st reviewer once again.

The possible reaction pathway of compound **14** (Path b):

The possible reaction pathway of compound **14** (Path a):

Scheme 3. The possible reaction mechanism for the product **14**

Yours Sincerely

Prof. Wei Zeng
 School of Chemistry and Chemical Engineering
 South China University of Technology
 Guangzhou 510641, P. R. China